# Genome-Wide Identification and Characterization of *Argonaute*, *Dicer-like* and *RNA-Dependent RNA Polymerase* Gene Families and Their Expression Analyses in *Fragaria* spp.

**DOI:** 10.3390/genes14010121

**Published:** 2023-01-01

**Authors:** Xiaotong Jing, Linlin Xu, Xinjia Huai, Hong Zhang, Fengli Zhao, Yushan Qiao

**Affiliations:** 1Laboratory of Fruit Crop Biotechnology, College of Horticulture, Nanjing Agricultural University, No. 1 Weigang, Nanjing 210095, China; 2Institute of Pomology, Jiangsu Academy of Agricultural Sciences/Jiangsu Key Laboratory for Horticultural Crop Genetic Improvement, Nanjing 210014, China

**Keywords:** strawberries, *AGO*, *DCL*, *RDR*, identification, evolution, expression

## Abstract

In the growth and development of plants, some non-coding small RNAs (sRNAs) not only mediate RNA interference at the post-transcriptional level, but also play an important regulatory role in chromatin modification at the transcriptional level. In these processes, the protein factors Argonaute (AGO), Dicer-like (DCL), and RNA-dependent RNA polymerase (RDR) play very important roles in the synthesis of sRNAs respectively. Though they have been identified in many plants, the information about these gene families in strawberry was poorly understood. In this study, using a genome-wide analysis and a phylogenetic approach, 13 *AGO*, six *DCL*, and nine *RDR* genes were identified in diploid strawberry *Fragaria vesca*. We also identified 33 *AGO*, 18 *DCL*, and 28 *RDR* genes in octoploid strawberry *Fragaria × ananassa*, studied the expression patterns of these genes in various tissues and developmental stages of strawberry, and researched the response of these genes to some hormones, finding that almost all genes respond to the five hormone stresses. This study is the first report of a genome-wide analysis of *AGO*, *DCL*, and *RDR* gene families in *Fragaria* spp., in which we provide basic genomic information and expression patterns for these genes. Additionally, this study provides a basis for further research on the functions of these genes and some evidence for the evolution between diploid and octoploid strawberries.

## 1. Introduction

RNA silencing plays a significant role in regulating development and maintaining the stability of the genome in response to biotic and abiotic stresses in plants [1]. RNA silencing via non-coding RNAs, such as microRNAs (miRNAs) and small interfering RNAs (siRNAs), can occur at the transcriptional, post-transcriptional, and translational stages of expression. Previously, diverse RNA silencing pathways in plants have been revealed by genetic and molecular analyses, including cytoplasmic sRNA silencing in the defense against viruses [2,3], silencing of endogenous messenger RNAs by miRNAs [4,5,6], and the siRNA-directed RNA-dependent DNA methylation (RdDM) pathway [7,8], which plays a huge regulatory role in the ripening of fruits [9,10] and the genomic imprinting of flowering plants [11]. These RNA silencing pathways are involved in several core components, including Argonaute (AGO), Dicer-like (DCL), and RNA-dependent RNA polymerase (RDR) proteins, etc. Firstly, RDR proteins catalyze the conversion of single-stranded RNAs (ssRNAs) into double-stranded RNAs (dsRNAs) [12,13]. Then, DCLs recognize dsRNA, and cleave it into 21–24 nucleotide (nt) sRNAs [13]. After cleavage and unwinding, single-stranded sRNAs are bound by AGOs, which form the cores of RNA-induced silencing complexes (RISCs) [14]. Finally, the RISCs act on the target RNA, resulting in either cleavage of the targets or repression of their translation. Alternatively, for the DNA targeting, AGO proteins are guided by the implementation of siRNA-directed RdDM machinery, affecting transcriptional repression [15].

The name of AGO is derived from the founding member of the *AGO1*, an allelic series of mutant of *Arabidopsis thaliana* that has narrow rosette leaves resembling the tentacles of a small Argonaute squid [16]. An ancient origin of AGO proteins has been reported in bacteria, archaea, and eukaryotes. For example, *Chlamydomonas reinhardtii* contains three AGOs [17], *Physcomitrella patens* has six [18], while the nematode worm *Caenorhabditis elegans* encodes 27 AGOs [19]. In the flowering plants, the AGO family expanded during evolution, with numerous duplications and losses. *A. thaliana* encodes ten AGO proteins [20], and the number of AGOs is even greater in other flowering plants. For example, there are 12 AGO proteins in pepper [21]; 13 in grape, sweet orange, and banana [22,23,24]; 15 in tomato and poplar [25,26]; 17 in maize [27]; 18 in rice and tea [28,29]; 19 in foxtail millet [30]; 21 in sugarcane [31]; 27 in tetraploid oilseed rape (*Brassica napus*) [32]; 28 in allotetraploid cotton (*Gossypium hirsutum*) [33]; and 69 *TaAGO* genes in the hexaploid bread wheat [34]. These proteins are ~100 kD and are shared with four domains, including ArgoN, PIWI-ARGONAUTE-ZWILLE (PAZ), ArgoMid, and a C-terminal PIWI domain [35,36]. Based on phylogenetic and functional analyses, these AGOs were classified into four different clades: AGO1/10, AGO5, AGO2/3/7, and AGO4/6/8/9 [37,38,39]. Among them, AGO1/10 clade is indispensable for the translational control of miRNA targets in regulating plant gene expression [40]. The AGO5 clade, which is sometimes classified as a subclade in the AGO1/10 clade [15,41,42], is also involved in the flower and seed developmental regulation [37,43]. Aside from this, AGO5 family member *MEL1* is found in rice-loading 21- nt reproductive phased siRNAs (phasiRNAs) [28]. Conversely, AGO2/3/7 shows close phylogenetic relationships but has been implicated in quite different function. For instance, AGO2 preferentially associates with endogenous sRNAs that have a 5′-terminal adenine [44] and is implicated in DNA double-strand break repair [45]. AGO2 and AGO3 arose from a duplication such that they are organized in a direct tandem [41]. However, AGO3 has differential biological functions compared with AGO2, which acts as an effector in the epigenetic pathway, but similar to AGO4 in that it predominantly recruits 24-nt sRNA and functions in RdDM [46]. AGO7 and AGO1 act together in trans-acting siRNA (tasiRNA) biogenesis [47], which is unique in that they preferentially associate with a single miRNA, miR390, to trigger the production of tasiRNAs [48]. Proteins of AGO4/6/8/9 have a high conservation consistent with their similar functions that preferentially associate with 24-nt heterochromatic siRNAs (hcsiRNAs) and begin with a 5′-terminal adenine to induce RdDM and cause epigenetic modifications. Then, gene expressions are altered at target loci [49,50,51].

Dicer proteins, a ubiquitous class of RNase III enzymes in plants, also named Dicer-like (DCL) enzymes, recognize and cut dsRNA to produce 21–24 nt sRNA [13]. Additionally, DCLs contain six domains: DEAD, Helicase-C, Dicer_dimer, PAZ, Ribonuclease_3, and the double-stranded RNA-binding domain (dsrm) [52]. Four *DCL* genes (*DCL1*-*4*) have been reported in *Arabidopsis* [52] and are involved in the biogenesis of distinct sRNA. Among them, *DCL1* is vital for miRNA biogenesis [53], *DCL2* generates 22-nt siRNAs [54,55], *DCL3* produces 24-nt siRNAs involved in RdDM and histone modification [56,57], and *DCL4* is essential for the production of 21-nt sRNA, including phasiRNAs and virus-activated siRNAs (vasiRNAs) [54,58,59,60]. Based on the above, DCLs have four classes, with more than one member for each class in the flowering plants. Five and eight *DCLs* have been reported in the grain crops, including rice [37], maize [27], and foxtail millet [30], respectively. For vegetable crops, seven *DCLs* have been identified in tomato [25], four in pepper [21], six in common bean [61], eight in *Brassica napus* [33], and eleven in allotetraploid cotton [62]. Moreover, *DCLs* were also found in the fruit crops, including grape (four *DCL*s) [22], sweet orange (five *DCL*s) [23], banana (three *DCL*s) [24], and peach (eight *DCL*s) [63].

As a copartner of AGO and DCL, RDR is involved in the processing of various siRNAs from dsRNAs in the RNA silencing and has a conserved domain: RNA-dependent RNA polymerase (RdRp) [64]. Thus far, *RDRs* have been identified in several flowering plant species, such as *Arabidopsis* [64,65], tomato [25], maize [27], rice [37], common bean [61], pepper [21], grape [22], sweet orange [23], pineapple [66], sugarcane [31], tea [29], banana [24], and poplar [67]. *Arabidopsis* comprises six *RDR* genes containing one each of *RDR1*, *RDR2*, and *RDR6*, as well as three *RDR3* subfamily members: *RDR3a/3b/3c* [64,65]. Among *RDRs*, *RDR1* in vasiRNAs biogenesis, and vasiRNAs-mediated antiviral defense, such as *Potato virus X* (PVX) [68], *Tobacco mosaic tobamovirus* (TMV) [69,70], *Tobacco rattle virus* (TRV) [69], and *Tomato leaf curl Gujarat virus* (ToLCGV) [71]. *RDR6* also participates in antiviral defense in plants, a high *RDR6*-dependent antiviral defense was maintained in the loss-of-function variant of *RDR1* in *Nicotiana* [72] and is precisely recruited to a cleavage fragment by 22-nt siRNA-containing AGO1-RISCs in coordination with SGS3 and SDE5 [73]. Moreover, the RDR6-SGS3-DCL4 siRNA system synergies with the flavonoid pathway regulate carbon metabolism fluxes in the genetic/epigenetic mechanisms of *Arabidopsis* [74]. *RDR2* plays roles in siRNA-mediated DNA methylation and histone modifications at telomeres for genome defense against transposons and viruses [13,75,76,77]. Currently, little is known about the function of the *RDR3* genes. Previous research revealed that *RDR3* helps to regulate hormone balance when plants compete with conspecifics in natural environments in *Nicotiana benthamiana* [78].

*Fragaria* are perennial herbaceous berry fruits, including a diversity of species with wild strawberries ranging from diploid (2n = 2x = 14) to decaploid (2n = 10x = 70), and a unique domestication of octoploid (2n = 8x = 56) cultivated strawberry (*Fragaria* × *ananassa*) [79,80]. Among these, wild diploid strawberry *Fragaria vesca*, with a small and sequenced genome (240Mb), is an excellent model for genetic transformation [81], and cultivated octoploid strawberry *F. × ananassa*, with an estimated genome size of 813.4 Mb, is an economically important fruit crop due to their alluring appearance, distinctive flavor, and health benefits [82]. Moreover, *F. vesca* is universally accepted as one of the diploid ancestors of the *F. × ananassa*, and the *F. vesca* subgenome has increased by retaining significantly more ancestral genes and a greater number of tandemly duplicated genes than other subgenomes in *F. × ananassa* [82,83,84,85,86,87,88]. Considering this, we performed a genome-wide analysis so as to identify *AGO*, *DCL,* and *RDR* gene families in diploid and octoploid strawberry. In order to comprehensively understand the function of these genes, we analyzed their protein structure, duplication events, evolutionary selections, and promoter cis-regulatory elements. Furthermore, the expression patterns of these genes were analyzed across various tissues during the development of strawberry. Finally, we examined the expression patterns of *AGO*, *DCL,* and *RDR* gene families in response to hormonal treatments. Above all, our study provides foundational genomic information for the *AGO*, *DCL*, and *RDR* gene families and its probable roles in strawberry growth and development.

## 2. Materials and Methods

### 2.1. Identification and Characterization of AGO, DCL and RDR Genes in Fragaria

The genome information and related annotation files of diploid strawberry *F. vesca* (v4.0.a2) [89] and octoploid strawberry *F. ×ananassa* (v1.0.a1) [80] were obtained from the Genome Database for Rosaceae [90]. Additionally, the protein sequences of all the AGO, DCL, and RDR genes of *A. thaliana* and *Oryza sativa* were downloaded from the TAIR (https://www.arabidopsis.org/ (accessed on 22 July 2020)) and Phytozome databases (https://phytozome.jgi.doe.gov (accessed on 22 July 2020)) and used to perform a BLAST (E-value 1e-10) search against strawberry genome sequences to obtain *Fragaria* orthologous genes (Appendix A). Meanwhile, hidden Markov models using HMMER 3.0 [91] software were employed for the identification of *DCL*, *AGO*, and *RDR* genes, respectively. Conserved domains of all candidate genes were analyzed and annotated using the Pfam database [92] and Conserved Domain Database in NCBI [93]. For the different transcripts of the same gene, the transcript with the longest open reading frame (ORF) is selected as the representative of the gene, and the transcript with the same length of ORF, the longest untranslated region (UTR) transcript is selected as the candidate gene for analysis. The physical and chemical parameters of the proteins were calculated using the ProtParam tool (https://web.expasy.org/protparam/ (accessed on 12 September 2020)).

### 2.2. Chromosomal Localization, Phylogenetic and Duplication Analysis

The images of all the candidate genes’ chromosomal locations were generated using MapGene2ChromosomeV2 (http://mg2c.iask.in/mg2c_v2.0/ (accessed on 20 September 2020)), according to the genome annotations of *F. vesca* and *F. ×ananassa*. The respective AGO, DCL, and RDR protein sequences were aligned using MEGA X [94], and phylogenetic trees were constructed using the neighbor-joining (NJ) method with the bootstrap test replicated 1000 times. Motif analysis was performed using MEME Suite [95]. TBtools was used to predict the segmental duplications and tandem duplications [96].

### 2.3. Ka/Ks and Cis-Regulatory Elements Analysis

DnaSP v6.12.03 was used to calculate the synonymous (Ks) and non-synonymous (Ka) substitution rates for homologous gene pairs [97]. The 2-kb region upstream of *AGO*, *DCL* and *RDR* genes was used for cis-regulatory elements analysis with PlantCARE (http://bioinformatics.psb.ugent.be/webtools/Plantcare/html/ (accessed on 30 December 2020)), and the results were illustrated using TBtools [96].

### 2.4. Hormone Treatments and Quantitative Real-Time PCR Analysis

*F. × ananassa* ‘Benihoppe’ plants were chosen for the study. The seedlings were sprayed with 0.2 mM naphthylacetic acid (NAA), 0.2 mM abscisic acid (ABA), 0.2 mM gibberellin 4 (GA_4_), 0.2 mM methyl jasmonate (MeJA), and 1.0 mM salicylic acid (SA). Additionally, the leaves of strawberry seedlings were collected after treating for 12, 24, and 48 h. All samples were rapidly frozen using liquid nitrogen and stored in a −80 °C freezer. Total RNA isolation and quantitative real-time PCR (RT-qPCR) were performed with reference to Zhang et al. [98]. *EF-1α* (XM_004307362) was used as the reference gene to normalize the expression level of target genes. The relative expression levels of genes were calculated using the 2^−ΔΔCt^ method. All the statistical analyses were performed using *t* tests. The primers for the quantitative real-time PCR are listed in Appendix A.

## 3. Results

### 3.1. Identification and Characterization of AGO, DCL and RDR Genes in Fragaria

Using the BLAST and HMMER search, we identified the putative *AGO*, *DCL* and *RDR* genes of *F. vesca* and *F. × ananassa*. The resulting sequences were further confirmed by analyzing conserved domains as putative family members according to the Pfam database [92] and Conserved Domain Database in NCBI [93]. As a result, we identified a total of 13 *AGO* genes, six *DCL* genes, and nine *RDR* genes in *F. vesca*, as well as 33 *AGO* genes, 18 *DCL* genes, and 28 *RDR* genes in *F. × ananassa*. Then, the candidate genes were named, and their basic information is listed in Appendix A.

Based on HMM research of PAZ (PF02170) and PIWI (PF02171) and BLASTp, 13 *AGO* genes were identified in diploid strawberry *F. vesca*, and we found that the lengths of *FvAGO* coding sequences varied from 2557 bp for *FvAGO5a* to 3381 bp for *FvAGO1b* and the code for proteins between 858 and 1126 amino acids. Additionally, most of them contain more than 19 exons, except for *FvAGO2*, *FvAGO7* (three exons), and *FvAGO4b* (one exon) (Appendix A). However, the lengths of the 33 identified *FaAGOs* greatly vary in octoploid strawberry. *FaAGO7d* has the smallest coding sequence of 1686bp and contains two exon regions, and *FaAGO1c* encodes the longest coding sequence of 6552bp containing 44 exons (Appendix A). The pI ranged from 8.6 to 9.7 and molecular weight was ~100 kDa in all the AGO proteins, except for *FaAGO1a*, *FaAGO1b* and *FaAGO1c*, where the molecular weight was more than 200 kDa. Similar to the foxtail millet [30], the protein instability index showed that most of the FvAGO/FaAGO proteins are unstable (Appendix A). A conserved domain analysis revealed that the ArgoL1, PAZ, and PIWI domains are present in all identified FvAGO and FaAGO proteins. The ArgoN domain is also present in all AGO proteins, except in octoploid strawberry, *FaAGO4b*, and *FaAGO7d*. Similarly, only *CsAGO5b* in *Citrus sinensis* lacks the ArgoL2 domain [23], and an *FvAGO5a* is also present in strawberry. Interestingly, we found that all AGO7 proteins also lack the ArgoL2 domain in diploid and octoploid strawberry. Moreover, the Gly-rich Ago1 domain was found in all AGO1 proteins, which is conserved in *AtAGO1*. Furthermore, we found that the ArgoMid domain is present in all the AGO1 and AGO10 proteins in *Fragaria*, and *FaAGO20* is the only domain starting and ending with an F-box domain and containing two FBA domains (Figure 1A).

The lengths of the 24 newly identified *DCL* genes coding sequences in *F. vesca* and *F. × ananassa* range from 3724 to 5895 bp with the coding potential for 1241~1964 amino acids, and the number of exons was greater than 19 (Appendix A). The molecular weight of FvDCL/FaDCL proteins varies from 147 kDa to 220 kDa and pI varies between 5.89 and 7.22. All the DCL proteins in *Fragaria* were unstable except FaDCL3e (Appendix A). All DCL proteins possessed DEAD, Helicase-C, Dicer_dimer (Duf283), PAZ, and Ribonuclease_3 (RNaseIII) domains, as reported for AtDCLs. In addition, *FaDCL2e* contains the RCRF and RF-1 domain (Figure 1B).

All nine *FvRDR* genes and 28 *FaRDR* genes found in the *F. vesca* and *F. × ananassa* genomes, code for proteins ranging between 587 and 2042 amino acids (Appendix A). The pI and molecular weight of these proteins range from 5.92 to 8.75 and 65 to 229 kDa, respectively (Appendix A). Compared with other FvRDRs in *F. vesca*, the FvRDR3 subfamily contains more exons with 22~25 exons (Appendix A). Interestingly, this is also true in *F. × ananassa. FaRDR3c* contains 47 exons and has the most exons and the longest DNA sequence with 19,000 bp in the RDR family in strawberry (Appendix A). Furthermore, a conserved domain analysis shows that *FaRDR3c* contains two RdRP domains, while all the RDRs in *Fragaria* share a common RdRP conserved domain (Figure 1C).

### 3.2. Chromosomal Localization of AGO, DCL, and RDR Genes in Fragaria

In order to comprehensively understand the evolution of multiple *AGO*s, *DCL*s, and *RDR*s, we analyzed their precise physical locations on the *F. vesca* and *F. × ananassa* chromosomes (Figure 2). The chromosomal location map shows that thirteen *FvAGO*s, six *FvDCL*s, and nine *FvRDR*s genes were unevenly distributed in all chromosomes. Five *FvAGO*s were detected on chromosome 4 (Fvb4), four on Fvb3, two on Fvb5, and one each on Fvb6 and Fvb7. Additionally, they were not localized on chromosomes 1 and 2. Three pairs of diploid strawberry *AGO*s, namely *FvAGO5a*-*FvAGO5b*, *FvAGO6a*-*FvAGO6b*, and *FvAGO1a*-*FvAGO1b*, were closely localized on Fvb3, Fvb4, and Fvb5, respectively. Therefore, we conjectured that they represent tandem duplications, as well as *FvRDR3a*-*FvRDR3b* on chromosome 1, *FvRDR6a*-*FvRDR6b* on chromosome 2 and *FvRDR1a*-*FvRDR1c*-*FvRDR1d* on chromosome 4. *FvDCL*s were unevenly distributed on chromosomes 1, 2, 6, and 7, and *FvRDR* genes were unevenly distributed on chromosomes 1, 2, 4, and 5. Chromosome 3 contains only members of the *AGO* family. Chromosome 7 contains nine genes and has the most gene members of the analyzed families (Figure 2A). The chromosome distribution of *FaAGO*/*FaDCL*/*FaRDR* genes families in octoploid strawberry is similar to that of diploid strawberry. The physical location map reveals the identified *FaAGO*, *FaDCL* and *FaRDR* genes on all the chromosomes of *F. × ananassa*, except for chromosome 7-4 that comprise no members of the analyzed families. Chromosomes 3-1, 3-2,3-3, 3-4, and 6-4 only contain the *AGO* genes. Chromosomes 1-1 and 7-3 have only one *DCL* gene each: *FaDCL4c* and *FaDCL1a*, respectively. Additionally, only *RDR* genes are distributed on 1-2 and 2-3. Importantly, *FaAGO6a* and *FaAGO15* appear very close on Fvb4-3, and the same applies to *FaAGO6b* and *FaAGO16* on Fvb4-1, as well as *FaAGO6e* and *FaAGO17* on Fvb4-4. For *FaRDR*s, most of the *FaRDR1* genes appear very close in the genome, such as *FaRDR1b* and *FaRDR1h* on chromosome 4-3 and *FaRDR1a*, *FaRDR1i*, and *FaRDR1j* on chromosome 4-4. Interestingly, all *FaRDR6* genes appear close together in the chromosomes except for *FaRDR6e*. *FaRDR6b* and *FaRDR6d* appear on Fvb2-1, *FaRDR6a* and *FaRDR6e* on Fvb2-2, and *FaRDR6c* and *FaRDR6g* on Fvb2-4 (Figure 2B).

### 3.3. MEME Analysis of AGO, DCL and RDR Genes in Fragaria

After going through the MEME program, we identified 10 of 20 conserved motifs in the AGOs from *Arabidopsis*, *F. vesca* and *F. × ananassa* (Figure 3A). Interestingly, half of the AGO proteins start with motifs 20 and 10, while the other half starts with motifs 20, 18, and 10, containing the motif 19, except for *FaAGO4b* and *FaAGO7d*. *FaAGO1a*, *FaAGO1b*, *FaAGO1c,* and *FaAGO5a* have two each of the motifs 5, 6, 8, 10, 11, 13, 17, and 20, which may be due to fragment duplication. In addition, *FvAGO6b*, *FaAGO15*, and *FaAGO16* contain two of motif 13. *FaAGO20* contains three of motif 20. For the DCL protein family, 15 conserved motifs were found in all DCL proteins (Figure 3B). Motif 19 is a mark that distinguishes the DCL1 subfamily from other subfamilies. In the DCL1 subfamily, motif19 is distributed at the end of the protein, and some DCL1 proteins contain two motif 19. *FvDCL2a*, *FaDCL2a*, *FaDCL2b*, and *FaDCL2b* contain two motif 14. *FaDCL2e* contains two motif 9. Compared with the DCL3 protein of *Arabidopsis*, the DCL3 protein of strawberry has one motif 10 and motif 12. In addition, we found that both the DCL1 subfamily and DCL4 subfamily contain motif 16. In the RDR protein family, the MEME analysis revealed eight distinct motifs, including motifs 8, 12, 6, 9, 3, 2, 1, and 5, which were identified as major motifs among the RDR family members (Figure 3C). However, the arrangement of these motifs in the RDR family is not the same. Motifs 12 and 9 had distinct diversification in the RDR3s proteins, different from all other RDRs. The above differences may lead to differences in the function of these proteins.

### 3.4. Phylogenetic Tree Analysis and Classification of AGO, DCL and RDR Genes in Fragaria

In order to examine the phylogenetic relationship of *AGO*, *DCL*, and *RDR* families, we constructed unrooted phylogenetic trees of all the identified AGO, DCL, and RDR protein sequences along with their *A. thaliana* homologs. FvAGO1a could not be included for the phylogenetic tree construction because the results of sequence alignment are unsatisfactory. The 12 FvAGO, 33 FaAGO, and 10 AtAGO proteins were subdivided into four clades, namely AGO1/10, AGO5, AGO2/3/7, and AGO4/6/8/9, following the nomenclature of the ten AGO genes of *Arabidopsis* (Figure 4A). The DCL family consists of four clades, including DCL1, DCL2, DCL3, and DCL4 (Figure 4B). Finally, the tree derived from RDR sequences also consists of four clades (RDR1, RDR2, RDR3, and RDR6) (Figure 4C).

### 3.5. Synteny and Ka/Ks Analysis of AGO, DCL and RDR Genes in Fragaria

Gene duplications plays an important role in the expansion and evolution of gene families. Gene duplications include whole-genome duplications (WGDs) and single-gene duplications. Single-gene duplications include tandem (TD), proximal (PD), and dispersed duplication (DD) [99]. WGDs are mainly responsible for genome evolution and genetic diversity in auto-polyploids [100], while segmental duplications and TDs are known to play an important role in the generation and maintenance of gene families [101]. PDs arise from ancient tandem duplicates interrupted by other genes or localized transposon activities [102]. DD denotes any DNA sequence non-locally duplicated in a genome, such as transposon element insertions and copy number variations [103]. The collinearity analysis of diploid strawberry and octoploid strawberry shows that, among the 107 identified genes in strawberry, 71 WGD events, 18 DD events, 10 TD events, and eight PD events were involved. Additionally, seven TD gene pairs were found: *FvAGO1a*-*FvAGO1b*, *FvAGO5a*-*FvAGO5b*, *FaAGO6b*-*FaAGO16*, *FvRDR3a*-*FvRDR3b*, *FvRDR6a*-*FvRDR6b*, *FaRDR6a*-*FaRDR6e*, and *FaRDR6b*-*FaRDR6d*. Notably, in the DCL family, all FvDCLs derived from DD duplication events, all FaDCL derived from WGD events, and no TD gene pair was found. Furthermore, we identified 77 (AGO 24, DCL 19, RDR 34) gene pairs arising from segment duplications in *F. vesca* and 55 (AGO 22, DCL 16, RDR 17) gene pairs arising from segment duplications between the *F. vesca* and *F. ×ananassa* (Figure 5). Then, Ka and Ks were calculated via DnaSP [97] using all orthologous and paralogous gene pairs CDs sequences identified via a synteny analysis. The findings show that most of the gene pairs exhibited Ka/Ks < 1, implying that the purifying selection may promote evolution. However, three orthologous gene pairs, *FvAGO6c*-*FaAGO14*, *FvDCL1*-*FaDCL1e*, *FvRDR1b*-*FaRDR1c*, and paralogous gene pairs, *FaAGO11*-*FaAGO14*, *FaDCL4b*-*FaDCL4c*, *FaRDR3b*-*FaRDR3e*, exhibited Ka/Ks > 1, suggesting a positive selection of these gene pairs (Appendix A).

### 3.6. Promoter Cis-Acting Element Prediction of AGO, DCL and RDR Genes

In order to further comprehend the cis-acting elements located upstream of identified genes, 2 kb sequences upstream from translations start sites of the three gene families in strawberry were analyzed. More than 16 cis-acting elements were identified in the promoters of the putative the three gene families in *F. vesca* and *F. ×ananassa* (Figure 6). These include hormone response elements, stress response elements, light-responsive elements, and tissue-development-related elements. Among the hormone-responsive elements, we found that 63.0% (29 out of 46) AGOs, 45.8% (11 out of 24) DCLs, and 54.0% (20 out of 37) RDRs promoter sequences carry the GA responsive element. Moreover, 41.3% (19 out of 46) AGOs, 25.0% (six out of 24) DCLs, 27.0% (10 out of 37) RDRs promoter sequences carry SA. Meanwhile, 80.4% (37 out of 46) AGOs, 83.3% (20 out of 24) DCLs and 62.2% (23 out of 37) RDRs promoter sequences carry ABA. Further, 78.3% (36 out of 46) AGOs, 75% (18 out of 24) DCLs, and 62.2% (23 out of 37) RDRs promoter sequences carry MeJA. Moreover, 39.1% (18 out of 46) AGOs, 66.7% (16/24) DCLs and 21.6% (eight out of 37) RDRs promoter sequences carry IAA. We also observed the presence of five other very important stress-responsive regulatory elements in our promoter sequences: defense and stress-responsive element, wound-responsive element, drought-responsive element, low-temperature-responsive element, and anaerobic induction element. In total, we identified the defense and stress-responsive element in 45 identified gene promoter sequences, drought-responsive element in 59 sequences, low-temperature-responsive element in 48 sequences, anaerobic induction element in 94 sequences, and wound-responsive element only in the *FvAGO6a* and *FvAGO6b* promoter sequences. All identified promoter sequences carry the light-responsive element. Additionally, some tissue-development-related elements were found, such as the meristem (29 sequences) and endosperm (21 sequences) expression elements, seed-specific regulation element (five sequences), zein metabolism regulation element (39 sequences), and circadian control element (11 sequences). Altogether, there are various types and numbers of regulatory elements that can provide vital evidence for the understanding of functions of *AGO*, *DCL*, and *RDR* genes in strawberry.

### 3.7. The Tissue-Specific Expression Patterns of AGO, DCL and RDR Genes in Diploid Strawberry

To analyze the functions of *AGO*, *DCL*, and *RDR* genes, we studied their expression pattern in the development of the flower tissues, seed tissues, and other tissues in diploid strawberry according to annotation of the *F. vesca* v4.0.a2 genome [89]. *FvAGO4a* were observed to be highly expressed in all tissues and predominantly expressed in embryo and anther12. *FvAGO1a* showed a high expression level in the style 2 stage and root. *FvAGO1b* was more highly expressed in the development of the wall than other genes. *FvAGO1b*, *FvAGO4a*, and *FvAGO5b* greatly accumulate in all developmental stages of cortex and pith. *FvAGO4b* and *FvAGO5c* presented very weak expression levels in all tissues except for pollen (Figure 7A). Additionally, we found that *FvAGO5b* had a higher expression level than *FvAGO5a* in the ghost, which had relations with the endosperm expression elements in the promoter of *FvAGO5b* (Figure 6). For the *FvDCL* family, all the *FvDCL* genes were weakly expressed in the pollen and embryo 3 stage. Tissue-specific higher expression of *FvDCL1* and *FvDCL3b* was observed in the anther 9 and 10 stages. Unlike *FvDCL3b*, *FvDCL1* was expressed at all stages of anther development. In style and seeding, all genes were weakly expressed, except for *FvDCL1*. Compared with the expression level of *FvDCL3a* in other tissues, it has the highest expression at anther78 stage. *FvDCL2a* shows a low expression level in all tissues (Figure 7B). Among *FvRDR*s, we found that *FvRDR1b*, *FvRDR1c*, and *FvRDR1d* all showed a very weak expression in the *FvRDR1* subfamily. In the *FvRDR3* subfamily and *FvRDR6* subfamily, the expression levels of *FvRDR3b* and *FvRDR6a* are very low. *FvRDR2* was significantly expressed during carpel development. Additionally, it showed a high expression in the flower meristem (FM), receptacle meristem (REM) and anther 78 stage. *FvRDR3a* has a higher expression level in the shoot apical meristem (SAM) (Figure 7C).

### 3.8. RT-qPCR Analysis of AGO, DCL and RDR Genes in Octoploid Strawberry Related to Hormones

Promoter cis-acting element prediction of *AGO*s, *DCL*s, and *RDR*s shows that five types of hormone-responsive elements were identified in their promoter regions, including ABA-responsive elements, and GA, MeJA, IAA, and SA-responsive elements (Figure 6). Based on this, the expression level of *AGO*, *DCL*, and *RDR* genes under five hormone stresses were analyzed by RT-qPCR. Herein, we found that almost all genes respond to all hormone stresses. Among them, *FaAGO6a* has a lower expression level in all hormone response (Figure 8). Moreover, we revealed that *FaAGO2* (ABA), *FaAGO7a* (MeJA), *FaAGO7b* (NAA), *FaAGO10a* (ABA), *FaAGO12* (ABA), *FaAGO14* (ABA and MeJA), *FaAGO18* (NAA), *FaDCL1d* (MeJA), *FaDCL2c* (MeJA and SA), *FaDCL3e* (NAA), *FaDCL4b* (GA), *FaRDR1d* (GA and MeJA), *FaRDR2e* (GA), and *FaRDR6d* (GA) have a clear response to one or two hormones (Figure 8, Appendix A). This finding is mostly consistent with the data of promoter cis-acting element prediction. For instance, three ABA-responsive elements were found in the promoters of *FaAGO2* and *FaAGO10a*, and two MeJA-responsive elements in the *FaAGO7a* and *FaRDR1d*. Auxin-responsive element was found in the *FaAGO7b*, and GA-responsive elements was found in the *FaDCL4b*, *FaRDR2e* and *FaRDR6d* (Figure 6). In addition, homologous genes have a similar expression pattern. For instance, *FaAGO15*, *FaAGO16*, and *FaAGO17* have a high expression level following 12 h of treatment (Figure 8). *FaDCL1* subfamily genes were also highly expressed after 12 h of treatment, except for *FaDCL1a*, *FaDCL4a,* and *FaDCL4b*, which show a clear response to GAs (Appendix A).

## 4. Discussion

### 4.1. AGO Proteins in Fragaria

In this study, the members of the AGO4/6/8/9 subfamily are the most common in diploid strawberry and octoploid strawberry, while *AGO3* is not found in diploid strawberry, and there is only one *AGO3* in octoploid strawberry. The expression profiling of the small RNAs revealed that *AGO4*, *AGO6*, and *AGO9* bind 24-nt siRNAs that derive from repeat and heterochromatic loci [49]. *AGO3* and *AGO2* are in the same subfamily, but the study found that the spectra of *AGO3*-associated sRNAs were different from those bound to *AGO2*, and *AGO3* could not complement the signature function of *AGO2* in host antiviral defense. Surprisingly, *AGO3* predominantly bound 24-nt sRNAs with 5′-terminal adenines, and the expression of *AGO3* driven by the *AGO4* promoter partially complemented *AGO4* function and rescued a DNA methylation defect in the *ago4-1* background, indicating that *AGO3*, similar to *AGO4*, is a component in the epigenetic pathway [46]. Therefore, we consider that this may be the reason for the low amount of *AGO3* in strawberries. *FvAGO4a* shows a higher expression level in embryo than ghost tissue, including endosperm and seed coat (Figure 7A), and our recent research revealed more 24-nt sRNAs present in the embryo than the endosperm [104]. These results show that 24-nt sRNAs recruited by AGO4 may play an indispensable regulatory role in the developing embryo and endosperm of strawberry. Recent studies found that these 24-nt siRNAs play an important regulatory role in the genetic molecular mechanism of endosperm genome imprinting [11]. Compared with *AtAGO7*, strawberry *AGO7* is missing the ArgoL2 domain. *FaAGO7d* and *FaAGO4b* are slightly shorter than other *AGO*s, and their domains are incomplete. We conjecture that they may be the product of octoploid strawberry genome-wide replication events. Next, we analyzed the characteristic DDH/H and DDD/H motifs in the AGO proteins of strawberry. However, *FaAGO4b* contains a DDH/P motif, *FaAGO7a* did not present this motif, suggesting that *FaAGO7a* may lack slicing activity (Appendix A). We also found that *FvAGO5a*, *FaAGO13*, *FaAGO15*, *FaAGO17*, and *FaAGO19* lack the slicing activity found among members of the strawberry AGO family. Two tandem duplication events *FvAGO1a* and *FvAGO1b*, *FvAGO5a* and *FvAGO5b* were found in chromosomes Fvb5 and Fvb3 in *F. vesca*. While the tissue-specific expression patterns suggested that these two gene pairs showed different expression regulation (Figure 7A). Two F-box domains and two F-box-associated domains, FBA_1 and FBA_3, were found in *FaAGO21*. In plants, many F-box proteins are represented in gene networks broadly regulated by microRNA-mediated gene silencing via the RNA interference [105], and these proteins play an important role in ubiquitin proteolysis machinery [106]. F-box proteins are also involved in many plants vegetative and reproduction growth and development. For example, F-box protein-FOA1 is involved ABA signaling to affects seed germination [107]. Interestingly, we found four ABA-responsive elements in *FaAGO21*. As a result, we concluded that *FaAGO21* may be significant in the development of strawberry.

### 4.2. DCL Proteins in Fragaria

Among the plethora of proteins involved in RNA silence or RNA interference (RNAi) by small RNA molecules, Dicer or DCL proteins are the primary key factors. The role of these RNaseIII-like enzymes is to excise 21–24 nt sRNA duplexes from structural dsRNA precursor molecules. In the *A. thaliana*, four DCL proteins mediate the production of various classes of sRNA. Additionally, in this study, six *FvDCLs* and eighteen *FaDCL* genes clustered into four subgroups were found in diploid and octoploid strawberry (Figure 4B). All the identified strawberry *DCL* genes contain the DEAD, Helicase-C, Dicer_dimer, PAZ, and two Ribonuclease_3 domains, whereas the *DCL1* and *DCL4* subfamilies had an additional dsrm domain, also known as dsRBD (double-stranded RNA-binding motif). The dsrm proteins are mainly involved in the post-transcriptional regulation by preventing the expression of proteins or mediating the localization of RNAs [108]. Interestingly, we found that motif 16 is conserved in the dsrm domain, which may be responsible for post-transcriptional regulation. *FvDCL1* and *FvDCL3* showed a higher expression level in flowers tissues than other genes, especially in the anther (Figure 7B). In pepper, *CaDCL1* and *CaDCL3* also exhibited a higher expression in flowers [21], and previous studies revealed that the double mutant of *dcl1* and *dcl3* exhibited a delay in the flowing of *Arabidopsis* [109]. Additionally, in peach, *PrupeDCL3* also showed a high level of expression in flowers [63]. These findings suggest that *DCL1* and *DCL3* play positive roles in the flower development of plants. In addition, *DCL2* and *DCL4* are key control virus replication levels, even in susceptible plants. Additionally, *DCL4* is also involved in the production of some miRNA, including miR822, miR839, and miR869 in *A. thaliana* [110], and plays important roles in the biogenesis of transposon-derived siRNAs that specifically target transposon (TE) transcripts and endogens, when TEs are epigenetically reactivated [111,112,113]. In this study, *FaDCL2e* showed two peptide chain release factors related to the RCRF and RF-1 domains, compared with other DCL2 proteins. In addition, *FaDCL2e* also has a highly significant expression pattern in hormonal treatments (Appendix A), which may have a special function in octoploid strawberry.

### 4.3. RDR Proteins in Fragaria

RDR is responsible for the synthesis of dsRNA on ssRNA substrates in either a primer-dependent or primer-independent manner [64]. The first plant RDR, *LeRDR*, was isolated in tomato [114]. In the *A. thaliana* genome, six RDR proteins were identified [64]. We identified nine RDR genes in the diploid strawberry, which is consistent with the number of RDR genes found in *Populus trichocarpa* [26]. Additionally, 28 RDR proteins were found in octoploid strawberry. The biological function of RDR proteins is usually linked to the subsequent action of specific DCL proteins. Plant *RDR1* is an important element of the RNA silencing pathway in plant defenses against viruses. *RDR1* expression can be elicited by viral infection and the exogenous application of salicylic acid (SA) and jasmonic acid (JA) [69,115]. Our results show that *FaRDR1d* and *FaRDR1g* are not only elicited by SA and MeJA, but also enhanced by GA (Appendix A). Additionally, *FaRDR1k* is also induced by ABA (Appendix A), which is similar to the *AcRDR1* [66]. *RDR6* also plays an important role in plant antivirus. However, unlike *RDR1*, the transcription of *NgRDR6* is induced by ABA, GA, JA, and CMV, but there is no significant change under the treatment of PVY, TMV, SA, and H_2_O_2_ [116]. Among the five members of the rice RDR family, only *OsRDR6* is induced by ABA and KINETIN but is not induced by SA, IAA, GA, and ethylene treatment [117]. In strawberry, *FaRDR6d* is significantly enhanced by GA treatment (Appendix A). *RDR2*, *DCL3,* and *AGO4* participate in siRNA-mediated DNA methylation to regulate gene expression. *RDR2* is mainly related to the epigenetic modification of plants. It participates in siRNA-directed DNA methylation to regulate gene expression together with *DCL3* and *DCL4*. In addition, *AtRDR2* plays a critical role in the development of the female gametophyte [118]. In diploid strawberry, we found one *RDR2* gene that was divided in the same clade as *AtRDR2*. *RDR2* is most highly expressed in the ovules of pineapple [66], while in diploid strawberry it has a high expression level in the development of carpel. Although they do not have the same organization, it was also suggested that these proteins may be involved in female reproductive development.

### 4.4. Evolution between Diploid F. vesca and Octoploid F. × ananassa

The genus of *Fragaria* contains approximately 25 species, diversifying ploidy, ranging from diploid (2n = 2x = 14) to decaploid (2n = 10x = 70) [78]. Among them, *F. × ananassa* is a modern cultivated species, an allo-octoploid (2n = 8x = 56), was derived from spontaneous hybrids between two wild allo-octoploid species *F. virginiana* and *F. chiloensis* in the 18th century in Europe [119]. However, there are still disputes about its diploid ancestors. For example, Tennessen et al. [81] clarified that *F. × ananassa* originated with four subgenomes, including *F. vesca*, *F. iinumae*, and two unknown ancestors, by presenting an approach Phylogenetics of Linkage-Map-Anchored Polyploid Subgenomes (POLiMAPS). However, Yang and Davis [82] reported that octoploid species of genetic signatures from at least five diploid ancestors, including *F. vesca*, *F. iinumae*, *F. bucharica*, *F. viridis*, and at least one additional allele contributor of unknown identity. With the chromosome-scale genome of *F. ×ananassa* assembled, Edger et al. [80] and Hardigan et al. [88] proposed that four species, including *F. vesca*, *F. iinumae*, *F. viridis*, and *F. nipponica*, contributed to the origin of octoploid strawberry. Moreover, Hardigan et al. [88] proposed the ABCD subgenome nomenclatures, and among them, *F. vesca* was the dominant source of the genic DNA, revealing that the chromosomes of *F. × ananassa*, Fvb1-4, Fvb2-2, Fvb3-4, Fvb4-3, Fvb5-1, Fvb6-1, and Fvb7-2 were the closest diploid *F. vesca* chromosomes. We found that these results were consistent with the result of gene location and synteny analysis in our study. For example, the number of *AGO*, *DCL,* and *RDR* in the *F. vesca* chromosomes was the same as those for the subgenome A of *F. × ananassa* except for Fvb3 and Fvb7 (Figure 2). Additionally, collinearity analysis showed that gene duplication events were discovered in the *F. vesca* and *F. ×ananassa* subgenome A (Figure 5).

## 5. Conclusions

In this study, a total of 13 *AGO*, six *DCL*, and nine *RDR*, and 33 *AGO*, 18 *DCL*, and 28 *RDR* genes were identified in diploid *F. vesca* and octoploid *F. × ananassa*, respectively. The conserved domains, motifs, and gene structures analyses showed the functional characterization of these gene families. Chromosome localization, phylogenetic tree, and collinearity analysis revealed that gene duplication events contributed to the evolution of these genes and provided some evidence of the evolution between diploid and octoploid strawberry. Moreover, their expression patterns in various tissues, developmental stages, and hormone stresses indicated the genes involved in the growth and development of strawberry. Therefore, our study provides a basis for further study on the functions of these genes and provides some evidence for the evolution between diploid and octoploid strawberries.

## Figures and Tables

**Figure 1 genes-14-00121-f001:**
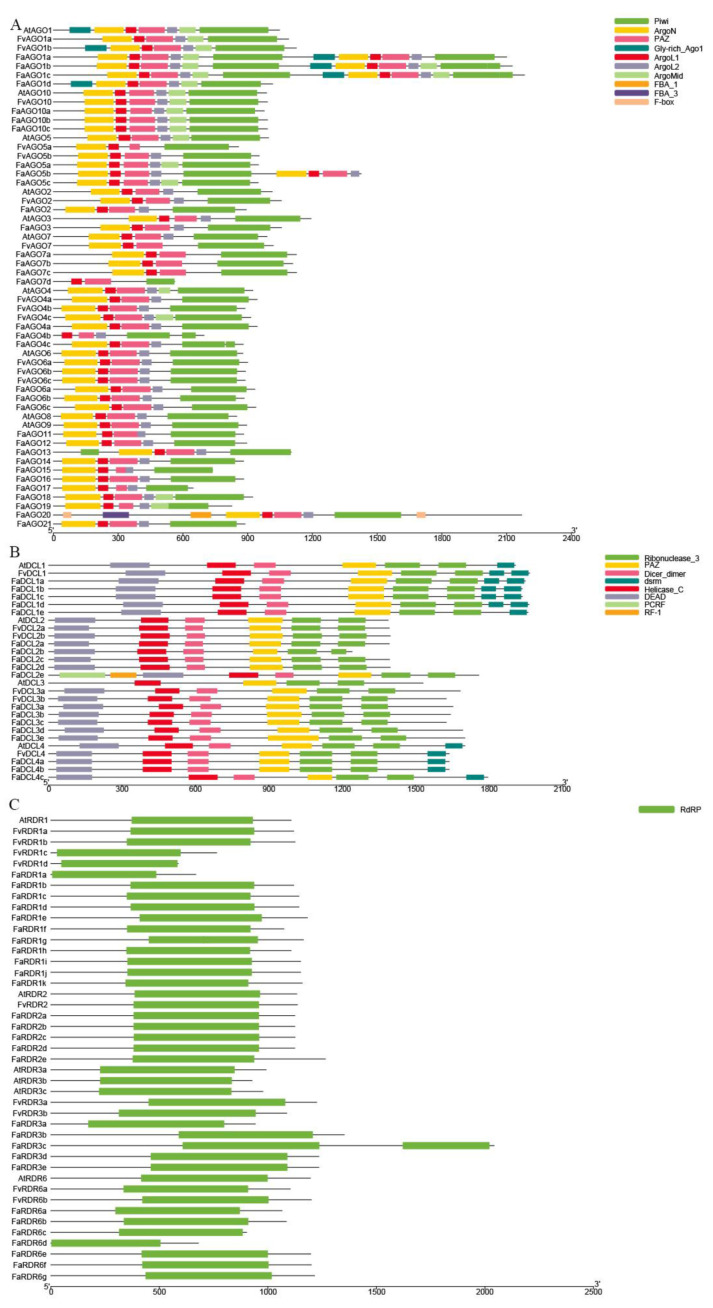
Domain analysis of AGO (**A**), DCL (**B**) and RDR (**C**) protein families of *F. vesca* and *F.* × *ananassa*. (**A**) ArgoL1, PAZ and PIWI domains are present in all identified FvAGO and FaAGO proteins. (**B**) All strawberry DCL proteins possess DEAD, Helicase-C, Dicer_dimer (Duf283), PAZ and Ribonuclease_3 (RNase III) domains. (**C**) All RDRs in *Fragaria* share a common RdRP conserved domain.

**Figure 2 genes-14-00121-f002:**
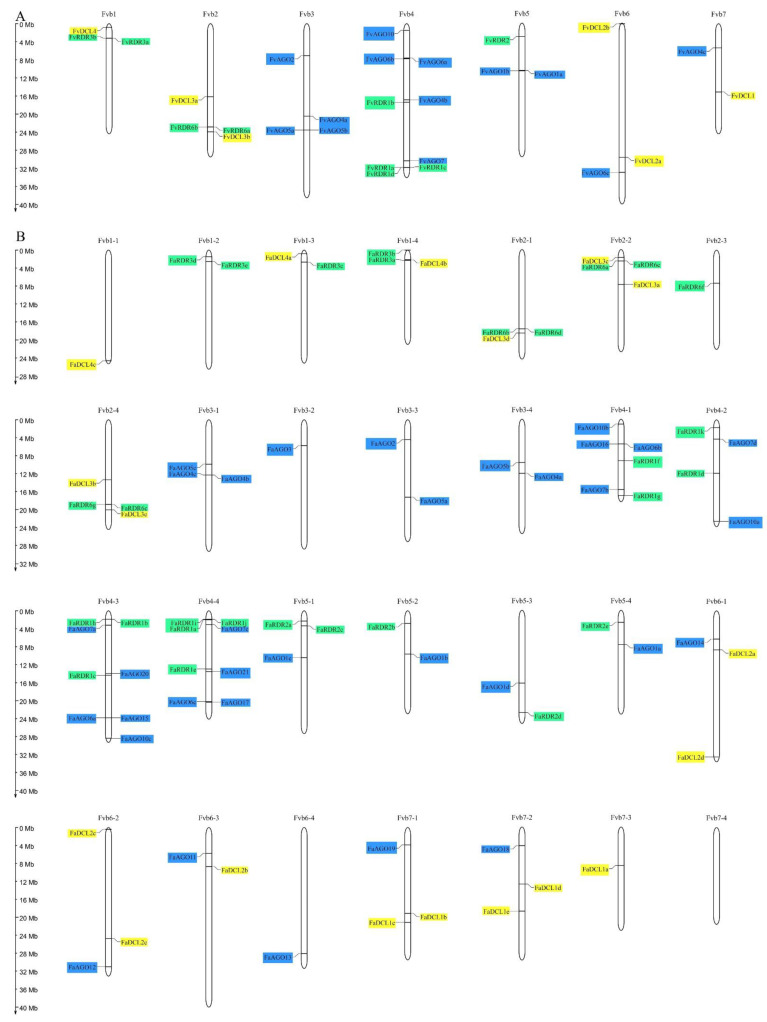
Chromosomal distributions of *AGO*, *DCL*, and *RDR* genes in *Fragaria.* (**A**) Precise physical location of *FvAGOs*, *FvDCLs*, and *FvRDRs* on *F. vesca* genome. (**B**) Chromosomal localization of *FaAGOs*, *FaDCLs* and *FaRDRs* on *F. × ananassa* genome.

**Figure 3 genes-14-00121-f003:**
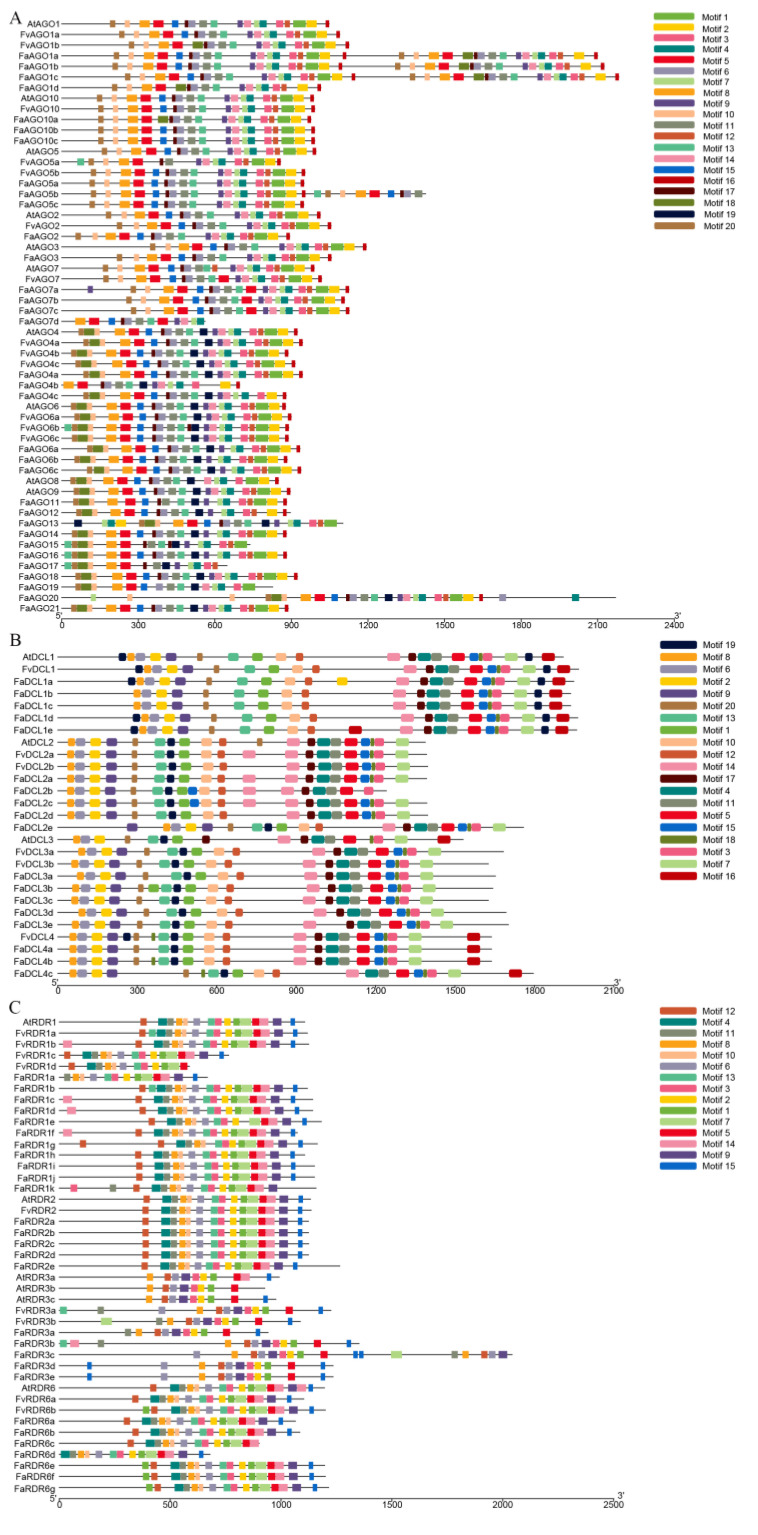
MEME analysis of strawberry AGO (**A**), DCL (**B**) and RDR (**C**) protein families. Motifs are represented by boxes with different colors, and box size indicates the length of motifs.

**Figure 4 genes-14-00121-f004:**
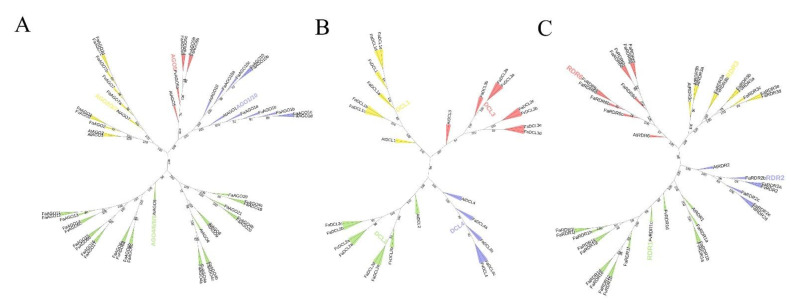
Unrooted neighbor-joining phylogenetic trees analysis of putative AGO (**A**), DCL (**B**) and RDR (**C**) proteins from *Arabidopsis* and *Fragaria*. Each gene family is divided into four groups shaded with a different color.

**Figure 5 genes-14-00121-f005:**
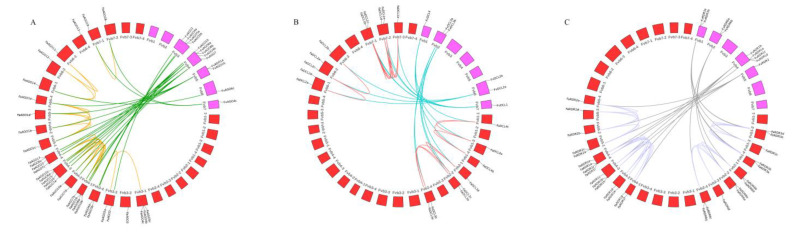
Chromosome distributions and synteny relationships of *AGO* (**A**), *DCL* (**B**), and *RDR* (**C**) genes in *F. vesca* and *F. ×ananassa*. The chromosomes of *F. vesca* and *F. × ananassa* are filled with purple and red, respectively. The duplicated gene pairs are connected with different color lines.

**Figure 6 genes-14-00121-f006:**
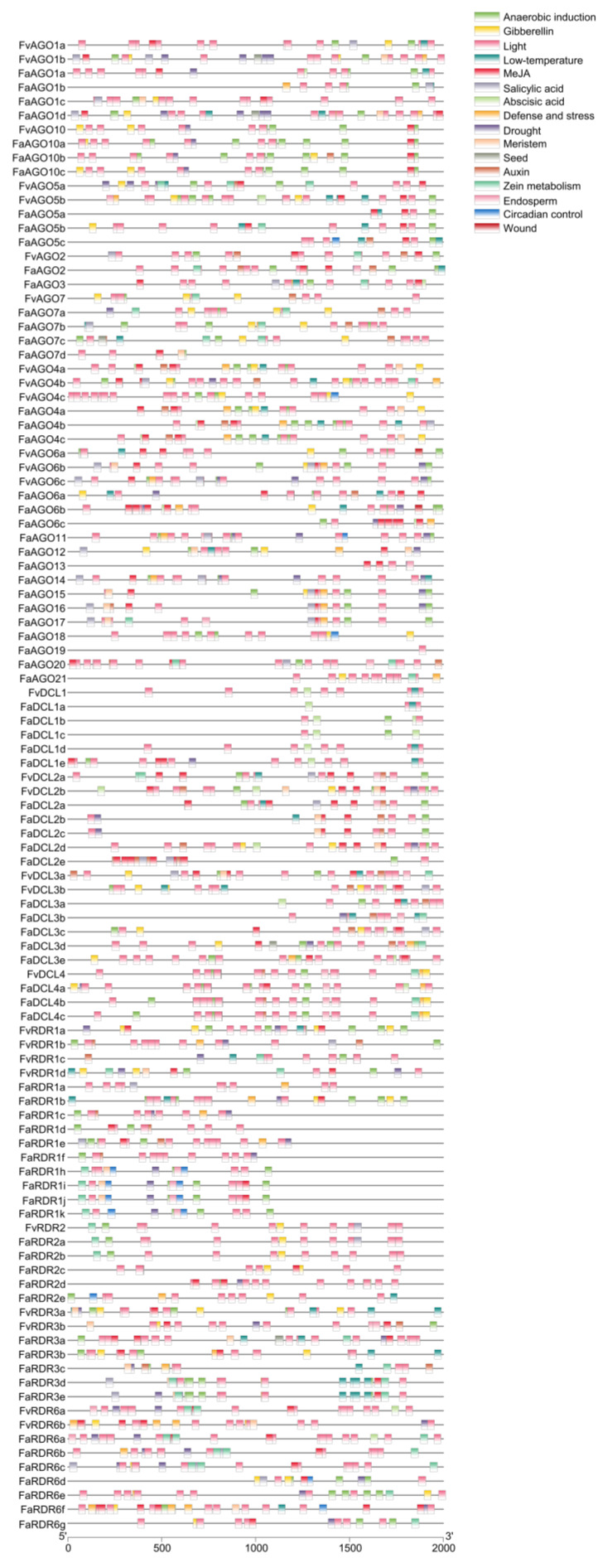
Promoter cis-acting elements were predicted in *AGO*, *DCL* and *RDR* gene families from *F. vesca* and *F. × ananassa.* Cis-acting elements are represented by boxes with different colors.

**Figure 7 genes-14-00121-f007:**
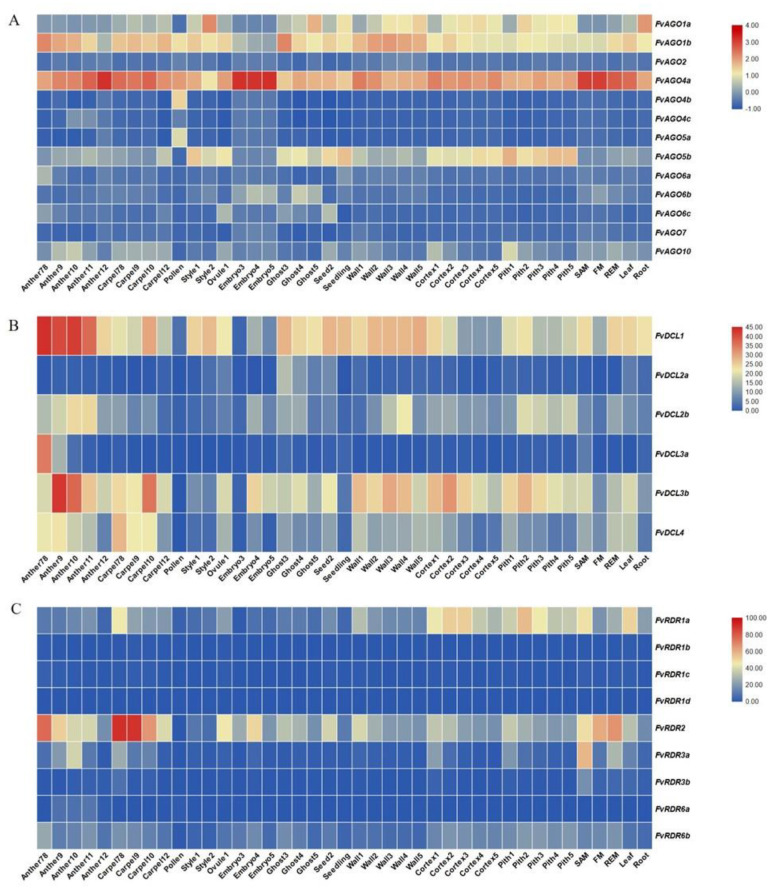
Expression profiles of *AGO* (**A**), *DCL* (**B**) and *RDR* (**C**) genes in diploid strawberry. Heatmaps showing the expression levels of *AGO*, *DCL* and *RDR* family members in different stages of flower tissues, seed tissues, and vegetative tissues. As shown in the bar to the right of map, gene transcript abundance is represented by different colors on the map. For the tissues of anther and carpel, 7, 8, 9, 10, 11, and 12 represent the developmental stage of *F. vesca* flower. For the other tissues including style, ovule, embryo, ghost, seed, wall, cortex, and pith, 1, 2, 3, 4, and 5 represent the developmental stage of *F. vesca* fruit.

**Figure 8 genes-14-00121-f008:**
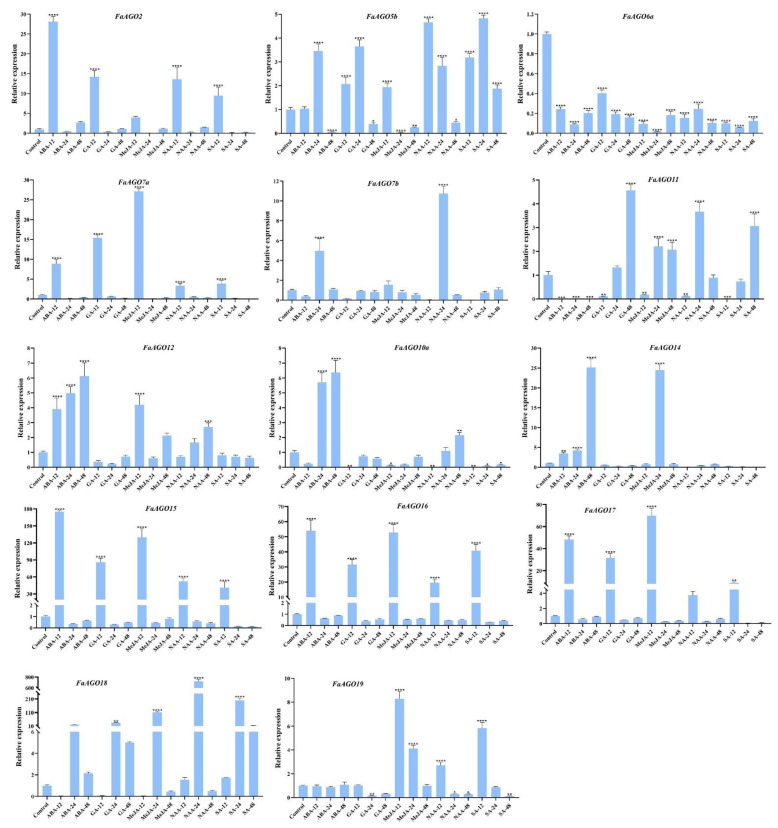
Relative expression profiles of *AGO* genes from cultivated strawberry *F. × ananassa* under five hormone treatments. Error bars represent standard deviation (SD). Columns with stars mark significant differences. The more stars, the higher the significant difference. All treatments are compared with “Control”, and * represents *p* ≤ 0.05, ** represents *p* ≤ 0.01, *** represents *p* ≤ 0.001, and **** represents *p* ≤ 0.0001. ABA-12, -24, -48: the leaves treated by ABA after 12, 24, and 48 h; GA-12, -24, -48: the leaves treated by GA after 12, 24, and 48 h; MeJA-12, -24, -48: the leaves treated by MeJA after 12, 24, and 48 h; NAA-12, -24, -48: the leaves treated by NAA after 12, 24, and 48 h; SA-12, -24, -48: the leaves treated by SA after 12, 24, and 48 h.

## Data Availability

The data presented in this study are available in the article and Appendix A.

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
