# Peer review of "Genome-Wide Identification and Characterization of Argonaute, Dicer-like and RNA-Dependent RNA Polymerase Gene Families and Their Expression Analyses in Fragaria spp."

_genes, 2023, doi:10.3390/genes14010121_

Round 1
Reviewer 1 Report
The present work is an extensive bioinformatic analysis of the genomes of two Fragaria species, the diploid wild Fragaria vesca and cultivated octaploid species Fragaria ananassa in the search for the homologs of the genes involved in RNA silencing in plants. RNA silencing mechanisms play an important role both in plant defence and development. It is the first study of this kind conducted for any Fragaria species. The study identified and characterized the genes and the encoded proteins belonging to the Ago, Dcl and Rdp families and formed a broad basis for the subsequent research into the precise role and function of these genes.
Introduction
Lines 62-64, the phrase is not clearly understandable.
Same for lines 87-89.
Line 121, using the term "perennial" may be suggested instead of "multi-year-old".
Materials and Methods
Seemingly the Methods section doesn't contain any reference to how the expression patterns for the AGO, DCL, and RDP genes identified in the study was performed. Was it a bioinformatic search, or the on the bench study of expression? In either case, the corresponding information should be added. Also in the Results section below, it would be useful to provide some more details about the developmental stages studied in Figure 7.
The Methods section also doesn't contain any indication as to how the qRT-PCR data was processed and which statistical criterion was used to confirm the significance of the differences observed. I strongly recommend adding this information.
Results
Line 184, there should be a typo, "BLSAT" should read as "BLAST"
The key to Figure 2 doesn't correspond to the actual figure. No A and B sections are indicated making it difficult to see which part of the figure corresponds to the Fragaria vesca genome and which to F. ananassa one. I would recommend to improve the figure.
Line 296, I would suggest adding some more details on MEGA limitations i this case.
Line 308, the "of" preposition might have been missed in the sentence.
Lines 403-406, the sentence requires rewriting.
Discussion
Lines 443, 452, I would recommend using phrases which are more common in scientific literature instead of "we think".
In the Result section the authors describe the identification of 19 motifs using the MEME software. However there is no further discussion of the possible role which these motifs play in the function of AGO, DCL and RDP proteins. What kind of motifs they are? Please, add some discussion on this point in the corresponding section.
I would recommend to the authors to reconsider the Discussion section in what regards the three gene families identified in the present work in the two representatives of the Fragaria genus. In the present form, it appears difficult to see how the data obtained in the work contribute to the current knowledge on the function and roles of these genes.
Reviewer 2 Report
The manuscript entitled "Genome-wide identification and characterization of Argonaute, 2 Dicer-like and RNA-dependent RNA polymerase gene families 3 and their expression analyses in Fragaria spp." constitutes its goals and novelty of the manuscript. The following minor suggestion/correction should be incorporated into the text.
1. In the abstract section, the author must define the response of identified genes to hormones.
2. Author can draw a rooted circular tree also.
3. Author must provide the detail (numbers) of the query sequence to search these genes.
4. Please provide the expanded form of abbreviated words and acronyms when using them for the first time.
5. The introduction is good. However, it is better for you to use more recent references in this field.
